# Evidence of Antimicrobial Resistance from Maternity Units and Labor Rooms: A Water, Sanitation, and Hygiene (WASH) Study from Gujarat, India

**DOI:** 10.3390/healthcare10040648

**Published:** 2022-03-30

**Authors:** Pachillu Kalpana, Poonam Trivedi, Priya Bhavsar, Krupali Patel, Sandul Yasobant, Deepak Saxena

**Affiliations:** 1Department of Public Health Science, Indian Institute of Public Health Gandhinagar (IIPHG), Gandhinagar 382042, India; kpachillu@iiphg.org (P.K.); ptrivedi@iiphg.org (P.T.); priyabhavsar@iiphg.org (P.B.); yasobant@iiphg.org (S.Y.); 2Parul Institute of Public Health, Parul University, Waghodia 391760, India; pkrups78@gmail.com; 3School of Epidemiology and Public Health, Datta Meghe Institute of Medical Sciences, Wardha 442004, India

**Keywords:** microbial contamination, antimicrobial resistance, WASH, maternity units, India

## Abstract

The main objective of this study was to determine the microbial contamination and antimicrobial resistance pattern among isolated bacteria from the environment surfaces of maternity units and labor rooms of healthcare facilities in the Gujarat state of India. The cross-sectional study was conducted in ten healthcare facilities, where the microbiological swab samples were collected from various pre-decided environmental surfaces of the maternity and labor rooms as part of the Water, Sanitation and Hygiene (WASH) assessment. The swabs were analyzed by conventional microbiological culture methods to identify microorganisms, including antimicrobial susceptibility testing. The study provides an insight into the microbial contamination of the visibly clean areas, i.e., the maternity ward, labor room, and general wards of the healthcare facilities. The labor rooms were found to be highly contaminated in comparison to other selected sites. The microbiological findings revealed a predominance of Gram-negative bacteria, specifically *Pseudomonas* species. The antibiotic susceptibility testing indicates resistance against many commonly used antibiotics. This study produces an identified necessity for enhancing microbiological surveillance in labor rooms and maternity units. This study also highlights the importance of microbiological status along with the WASH status of healthcare facilities.

## 1. Introduction

The average annual maternal mortality rate of 2.9 percent is substantively less than half of the 6.4 percent annual rate needed to achieve the Sustainable Development global goal of 70 maternal deaths per 100,000 live births [1]. Almost 94% of all maternal deaths occur in low- and lower–middle-income countries (LMICs). One of the influential groups of women among them is that of adolescent girls under 15 years of age who are usually at risk of mortality due to complications during and following pregnancy and childbirth, also the leading cause of maternal death in LMICs [2,3].

Critical steps are required for further advances in reducing maternal deaths and understanding the causes of deaths among mothers. Hemorrhage and sepsis are the main causes of maternal mortality, accounting for nearly 43% [4]. The evidence provides links between poor Water, Sanitation, and Hygiene (WASH) practices and the role of the environment at the time of birth contributing to life-threatening infections in mothers and newborns [5]. The healthcare facilities are standardly categorized as ‘clean’ based on visual assessment, which is subjective and inappropriate, and such an assessment does not correlate with the microbiological risks [6]. The presence of these microorganisms, including human pathogens, which are invisible to the naked eye, facilitates the spread of infection in the hospital environment [7]. The infections that occur in a health care facility while receiving health care, appear 48 h or more after hospital admission, or within 30 days of having received health care are known as health care-associated infections (HCAIs). The numbers show that out of every 100 patients hospitalized, seven patients in developed countries and ten patients in developing countries become infected with HCAIs [3]. The consequences of HCAIs relate to increased morbidity, mortality, length of hospital stays of patients, and cost of treatment. The annual expenses of HCAIs alone in the USA are between USD 28 and USD 45 billion, but even with this amount of spending, 90,000 lives are still lost [8].

Sepsis accounts for the cause of 10–15% of deaths among pregnant or recently-delivered women [9]. The risk of maternal death related to an infection at birth or after the delivery has been known for centuries [10]. Whether a potential pathogen causes infection depends on many factors, including the host [11]. The patients on maternity units, both mothers and newborns, face multiple risks due to the physiological processes of birth, such as cutting the umbilical cord, perineal tears, or cesarean section wounds. Furthermore, the prevalence of HCAIs reflects multiple dimensions, such as missed opportunities for prevention and the vast majority of infections occurring in LMICs.

One of the other increasing threats with HCAIs is their increasing antimicrobial resistance (AMR), which had prominence at the 71st United Nations General Assembly (UNGA) and captured global attention. Specifically, in LMICs where the lack of regulations on prescribing antibiotics and prophylactic use is slipping into routine use for all types of deliveries by healthcare workers. This scenario partly represents the healthcare workers’ recognition of the inadequate state of hygiene in healthcare facilities and, partly, assumptions about the poor personal hygiene of delivered women [12]. However, there is a comparative neglect of monitoring for AMR threats in maternity units’ crucial environment, which can elevate the issue of HCAIs and related consequences.

The role of clean water and adequate sanitation and hygiene to control and prevent infections in healthcare settings has been studied [13]. However, a dearth of studies is found in the case of maternity units on the topic of AMR. Thus, in the present study, we have explored the presence of microbial contamination and antibiotics resistance in maternity units, which will provide crucial evidence for the labor room and maternity wards’ environment cleanliness. The study also tried to identify the major susceptible sites for the infections and which pathogens mainly contribute to microbiological contamination and the antibiotic resistance pattern in the maternity wards and labor rooms of healthcare facilities of Gujarat, India.

## 2. Materials and Methods

The present study involved maternity units from a total of ten Health Care Facilities (HCFs) consisting of six Primary Health Centers (PHCs) and four Community Health Centers (CHCs) of the selected districts in Gujarat, India. The Health Care Facilities for the study were selected based on the following criteria: (a) in PHC at least 15 deliveries per month, (b) in CHC at least 30 deliveries per month, and (c) availability of adequate health staff (more than 75% filled up staff). A round table was conducted among experts from diverse fields, which consisted of microbiologists, WASH experts from United Nations Children’s Emergency Fund (UNICEF), State Infection and Prevention Control Department, Government of Gujarat, Ahmedabad Municipal Secretary, Indian Association of Preventive and Social Medicine (IAPSM), and GMERS College, Gandhinagar for the selection of the microbiological sample collection sites in HCFs. The complete study included a need assessment, walk-through survey, and microbiological surveillance of all health care facilities. The trained research assistants administered a structured questionnaire in vernacular language for the need assessment and walk-through survey. The microbiological samples were collected for the selected sample collection sites from maternity units and a few from the general ward in HCFs. The chosen sites mostly included bed, mops, sink-taps, instruments, buckets, NBCU side area, table, toilet, and water point.


**Sample collection**


The swab sampling method was used to collect samples for microbiological surveillance. The sterile cotton swab was dipped in 5 mL of sterile water and rubbed against the selected test surfaces by holding the swab flat against the surface and applying equal pressure. The swabs were placed back in a test tube case using an aseptic procedure and labeled. The samples were transported through a cold chain at 2 to 8 °C within 2 h of collection.


**Isolation and identification of bacteria**


The different culture media selective and non-selective media such as nutrient agar, chocolate agar, and MacConkey agar were prepared according to the manufacturer’s instructions and distributed on the Petri plates. All the swab samples were spread on nutrient agar, chocolate agar, and MacConkey agar and incubated for 24 h at 37 °C. After incubation, typical colonies for each bacterial group were evaluated, identified, and classified by standard microbiological techniques. For the qualitative analysis of the presence or absence of organisms, samples were tested through Gram staining, catalase, coagulase, mannitol salt agar (MSA), oxidase, indole, Voges–Proskauer (VP) test, citrate, sugar fermentation tests, decarboxylation, and Triple Sugar Iron test (TSI).


**Antibiotic Susceptibility testing**


Thirty-six strains were tested for antibiotic resistance by the standard agar disc diffusion technique on Muller Hinton agar using commercial discs (Himedia, India). All the found microorganisms were tested for antibiotic sensitivity, including Gram-positive and Gram-negative. According to the Clinical Laboratory Standard Institute (CLSI) guidelines, antibiotics were used for each found microorganism.

## 3. Results

In total, bacterial growth was observed in 36 cultures (18.46%) out of 195 swab samples collected from different selected sites of the maternity units, labor room, and general ward of the selected 10 HCFs.

The most prevalent Gram-negative bacteria were *Pseudomonas aeruginosa* with 38.9% positivity, an important nosocomial pathogen associated with hospital infections and reported for the association of this organism with puerperal septicemia. *Pseudomonas* species was represented with 27.8% followed by 11.1% for *Acinetobacter* species. On the other hand, *E. coli* represented only 5.6% of total positive samples taken, as well as *Klebsiella* species also with 5.6%, while the most prevalent Gram-positive bacteria were *Staphylococcus* species with 11% positivity. *Pseudomonas aeruginosa* were obtained by a simplified methodology. Some of the microorganisms could not be identified at the species level. These microbiological findings revealed an extremely high percentage of contamination with *Pseudomonas* species, which comprises mainly *Pseudomonas aeruginosa* and other species of *Pseudomonas*, which were not further classified. Overall, there was a predominance of Gram-negative microorganisms.

Comparison between the microorganisms present or absent according to the sample sites is described in Table 1. The most contaminated sites were found to be mops followed by beds, sink-taps, and buckets of maternity wards, labor rooms, and delivery rooms. In addition, a few of the microorganisms were found in the staff toilet, mops, and taps of general wards.

The labor rooms were found to be highly contaminated with 58% of the total positive samples found across all selected sites. This was followed by the contamination of the maternity ward with 27.8% of the total samples, then the general ward with 13.9%, and the delivery room with 0%. Antibiotics sensitivity tests were further conducted for all the sample sites’ isolated microorganisms using CLSI guidelines. The microorganisms exhibited divergent results on the basis of the genus of the bacteria and the used antibiotics. Table 2 shows the antibiotic sensitivity test of the isolated bacteria to sample sites.

### 3.1. Pseudomonas aeruginosa

For the antibiotic susceptibility test on *Pseudomonas aeruginosa* from the maternity ward and labor room, swab samples of the mops showed about 50% resistance against the antibiotic. On the other hand, this was found to be 47.1% for the resistance of *Pseudomonas aeruginosa* against antibiotics in the case of isolates found from bucket swab samples, whereas, in the case of swab samples collected from the general ward, *Pseudomonas aeruginosa* was only susceptible to the same antibiotics. It was resistant to antibiotics ceftazidime, ticarcillin-clavulanate, cefepime, and ciprofloxacin and was sensitive towards all other tested antibiotics.

### 3.2. Pseudomonas Species

Similar antibiotic sensitivity tests were repeated for all the isolated pathogens. In the case of *Pseudomonas* species found from the maternity wards’ and labor rooms’, mops, beds, and sink-taps, samples showed 38.1%, 28.6%, and 21.4% resistance, respectively. This species was found to be resistant to ceftazidime, ticarcillin-clavulanate, cefepime, ciprofloxacin, and levofloxacin antibiotics.

### 3.3. Acinetobacter Species

*Acinetobacter* species were found from the bed and sink-tap swab samples of the labor room. The antibiotic sensitivity spectrum of this species showed sensitivity towards all the examined antibiotics.

### 3.4. E. coli

*E. coli* was found in swab samples of the sink-tap in the labor room and swab samples of the mop from the general ward. All the isolates of *E. coli* revealed sensitivity to all the antibiotics used and thus have a susceptible antibiotic spectrum.

### 3.5. Klebsiella Species

The *Klebsiella* species were found from the swab samples of the beds and sink-taps of the maternity wards/labor rooms. The isolate was examined for the antibiotic resistance pattern, and the results showed that the isolate from the bed swab and sink-tap samples was about 18.2% resistant. It was found to be resistant to antibiotics ampicillin, cefazolin, and cefuroxime. However, the two genera of *Klebsiella* exhibited different responses exposed to antibiotics ciprofloxacin and cotrimoxazole.

### 3.6. Staphylococcus Species

The *Staphylococcus* species were identified in the swab samples of the beds and sink-taps of the labor and maternity rooms. The results showed that 50% of the bed samples and 43.8% of the sink-tap samples were resistant. Furthermore, 81.3% of the table samples were also resistant. They were resistant to the antibiotic penicillin, and all the genera exhibited different responses such as being resistant and sensitive to antibiotics amoxicillin-clavulanate, oxacillin, cefoxitin, erythromycin, azithromycin, clarithromycin, clindamycin, cotrimoxazole, chloramphenicol, ciprofloxacin, levofloxacin, and gentamicin. All isolates of *staphylococcus* were sensitive towards antibiotics linezolid, tetracycline, and vancomycin.

## 4. Discussion

The proportion of institutional deliveries in LMICs has increased significantly, representing an indicator of success in improving the quality care of mothers during childbirth [14]. This high uptake of maternity care services parallelly demands the importance of enhancing the quality of care in HCFs [12]. A lack of quality care can upturn this situation and contribute to morbidity and mortality. The direct causes of maternal death are obstetric hemorrhage, hypertensive disorders of pregnancy, puerperal sepsis, and abortion-related deaths, constituting approximately 50% of all maternal deaths [15]. A significant proportion of the direct causes of maternal death are considered preventable, but puerperal sepsis, pregnancy-related infection, contributes to the highest case fatality [15]. In India, a clean birthing practices campaign, training programs for skilled birth attendants, and Janani Suraksha Yojana to promote institutional deliveries played a crucial role in reducing the puerperal sepsis rate [15]. The introduction of antibiotics added to a reduction in the sepsis rate. However, it was not, of course, due solely to antibiotics. The prominent role of hygiene and cleanliness of the birthing environment in reducing pregnancy-related infection has been reported multiple times [6,7,16,17]. The global focus of hygiene majorly concentrates on hand hygiene in preventing HCAIs. Hand hygiene plays a much more critical role, but the requirement of a hygienic physical environment needs to accompany this to break the transmission chain of infection [8]. Visual cleanliness taken as a proxy for safety does not provide microbiological safety [18].

This study’s primary focus is on reporting the microbiological contamination of the labor rooms and maternity wards. In the present analysis, 195 surface samples with a microbiological contamination rate of 18.46% (*n* = 35) were obtained. The microbiological surveillance data on the maternity ward and labor room from India for comparison purposes are lacking. Some other studies from critical areas of HCFs such as operation theatres and intensive care units of HCFs reported positivity rates ranging from 8.24% to 45.8% [19,20]. The general ward surveillance has also been included in the study because of the increase in the lack of available facilities for institutional deliveries leading to the use of general ward beds for delivered mothers in some HCFs. The highest positive microbiological samples were obtained from the mops used to clean labor rooms, delivery rooms, and maternity wards. Based on the results and the available literature, the strong case to argue against using visual cleanliness to determine safety in terms of the presence of potential pathogens and the chain of transmission of pathogens remains clear. The focus on the infection prevention control program has been encouraged, but, on the other side, the different aspects of infection control need to be accounted for too. We want to highlight that the contamination of cleaning tools such as mops is crucial and requires urgent attention for a better mechanism to be implemented to prevent further spread of pathogens and infection in HCFs. The unavailability of any regulations regarding mop use and the time period for changing mops in HCFs also needs to be considered. Cleaners should also be an integral part of the infection control team [19].

The important impact of a hygienic environment on the prevention of transmitting infections and aiding in the recovery of patients has been discussed [20]. The swab sample profiles of the bed areas of labor rooms, maternity wards, and delivery rooms showed these were the second most contaminated areas in this study. The microbial contamination of these areas can be due to various reasons, from the considerable presence of amniotic fluid, blood samples, tissues, and other biological fluids that serve to be the breeding ground for microorganisms’ source and spread [21]. Patients admitted in these special wards are much more prone to infections, and, thus, the amplified microbial pathogens may pose a serious threat to the health of the mother as well as a newborn with life-threatening consequences [22].

The analysis in this study revealed the influential presence of Gram-negative pathogens *Pseudomonas aeruginosa*, *Pseudomonas* species, *Acinetobacter* species, *E. coli*, and *Klebsiella* species to be more dominant as compared to Gram-positive pathogens such as *Staphylococcus* species. The scenarios become much more chaotic with the development of resistance of a growing number of microorganisms. The antibiotic susceptibility test results of the microorganisms found in the study represent both resistance and multiple resistance to antimicrobials. The resistance has escalated to more than three antibiotics in some of the microorganisms, which creates the issue of treatment failure. This stresses the importance of the development of microbiological surveillance and focuses on the serious gaps in knowledge on antimicrobial resistance.

*Staphylococcus aureus (S. aureus)* infections are on the list of most common and problematic HCAIs [23]. *S. aureus* has been reported from the surgical room of HCFs and has been shown to be the cause of frequent HCAIs happening at a hospital [24]. A double outbreak of *S. aureus* was reported and was the cause of staphylococcal scalded skin syndrome (SSSS) as a hospital-acquired infection in the maternity unit [25]. The surveillance report then reported the probable sources of *S. aureus* from the hospital and was epidemiologically linked to cases of SSSS. A control measure was suggested in this report, which includes the use of chlorhexidine-containing detergent, and this can be adapted at all HCFs for prevention against *Staphylococcus* infections. The continual trend of *S. aureus* gaining resistance against penicillin in the first decade of its introduction, and the development of methicillin-resistant *S. aureus* (MRSA) are adding threats to the available treatment options [26].

*Pseudomonas aeruginosa* is considered one of the most common Gram-negative bacteria causing nosocomial diseases and HCAIs, particularly among hospitalized patients [27]. The pathogen is responsible for severe infections, mainly in immunocompromised patients, causing bacteremia, complicated intraabdominal or urinary tract infections, and ventilator-associated pneumonia. *Pseudomonas* species are a few water-based pathogens, also known as opportunistic premise plumbing pathogens (OPPPs) [28]. In healthcare settings, a moist environment can potentially serve as reservoirs for these water-based microorganisms [29]. The microbiological findings of this study also align with the presence of *Pseudomonas* species predominately found around water-based surfaces such as a bucket, sink-tap, and mops. The presence of these microorganisms on high-touch surfaces such as taps, thus represents the source of transmission via multiple routes to the patients and can result in life-threatening infections. The *Pseudomonas* species and *Pseudomonas aeruginosa* need to be focused upon because of their natural resistant ability against certain antimicrobial drugs and as they are highly prone to acquire resistance to many currently used antibiotics [30]. This multitude of resistance mechanisms presented by *Pseudomonas* species narrows down the therapeutic window, making the powerful antibiotics useless.

*Acinetobacter* species are responsible for nosocomial infections and can cause severe pneumonia and infections of the urinary tracts, bloodstream, and some other parts of the body. The primary source of *Acinetobacter* is hospitalized patients with lower immune defenses infected by Gram-negative microorganisms. The survival rate of *Acinetobacter* species on surfaces in hospitals is relatively high, and the attack mechanism involves wounds and invasive devices [31]. The microbiological surveillance in the present study represents the *Acinetobacter* species found in the samples collected from the bed areas of the labor room, maternity ward, and delivery room. The environment plays a significant role in the transmission of infection, and the survival of *Acinetobacter* in dry environments for weeks enables their transmission through contaminants in hospitals [31]. A study reported one of the species of *Acinetobacter*, *A. baumannii*, in a Turkish University Hospital that showed a level of 16.6% of nosocomial infections being due to *A. baumannii*, with the majority of positivity found from the intensive care unit of the hospital [32]. *A. baumannii* is considered one of the most common species widely present in the hospital environment [33]. This bacterium can infect, cause fatal illness, and increase patient mortality and hospital costs. Additionally, *A. baumannii* is known for its resistance to most antibiotics, a major public health emergency. Environmental sources and asymptomatic carriers of various microbial pathogens are responsible for the majority of the outbreaks. Transmission can occur through direct contact or indirect contact from environmental surfaces via contaminated hands. *E. coli* and *Klebsiella* species may survive desiccation for more than a year. Hospital sinks were common reservoirs for multi-drug resistant organisms and *E. coli* and *Klebsiella* species in this present study [34]. A recent study reports the dominance of isolates of *Klebsiella pneumoniae* and *E. coli* in most of urinary tract infection (UTI) patients among pregnant women [35]. UTI in pregnancy is associated with complications to the mother and fetus before and after delivery such as pyelonephritis, sepsis, septic shock, anemia, acute and chronic renal failure, premature delivery, and fetal mortality. Antimicrobial resistance is a major health problem in the treatment of UTIs [36].

WHO defines infection, prevention, and control (IPC) as a scientific approach and practical solution designed to prevent harm caused by infection to patients and health workers [37]. IPC holds an integral position in providing safe, effective, high-quality health service delivery and universal health coverage. Effective IPC plays a crucial role in preventing HCAIs and can reduce HCAI rates by 30% [38]. The intersection of IPC with WASH, including waste management and environmental cleaning, cannot be missed. IPC cannot be met without WASH. The lack of an adequate infrastructure, including poor WASH services, provides the basis for inadequate IPC in HCFs. An increase in the influx of patients in HCFs can also be impacted by the shortage of a dedicated healthcare workforce, lack of IPC training, and inadequate microbiological surveillance in providing quality care to patients [39,40]. Poor WASH and IPC result in HCAIs, the transmission of diseases, and the increased use of antibiotics exacerbates outbreaks and the spread of infections.

## 5. Conclusions

Healthcare-associated infections and their link to the presence of microbial pathogens with a significant focus towards intensive care units and operation theaters has been reported worldwide. There is still a lack of data specifically from the labor room and maternity ward. Our study reports data on the microbiological contamination of the labor room, maternity ward, and delivery room of selected HCFs in Gujarat. The found microorganisms can cause infections in delivered mothers and newborns when connected with the reported literature. It documents the importance of securely ensuring that disease and harmful microorganisms are contained. Priority towards the microbial safety of HCFs should be more focused rather than the visual cleanliness proxy of hygiene considered up to now. Microbiological surveillance should be an essential part of the infection control program and WASH practices in HCFs. The need for an agreed and standard method for sampling healthcare surfaces, frequency of sampling, and acceptable levels of surface contamination in the health care system is enhanced. Microbiological surveillance along with surveillance of antimicrobial resistance patterns will aid in identifying the trends of infection and resistance in advance. The increasing institution delivery rates in Gujarat mean that it needs to be stressed that the hospital environment cannot change the clinical outcomes unless the quality of care is more focused. The surveillance models and microbiological data need to be improved in all healthcare facilities to monitor microbial contamination in labor rooms and maternity wards to provide a safe clinical environment for the mother and newborn.

## Figures and Tables

**Table 1 healthcare-10-00648-t001:** Presence of microorganisms stratified with the sample sites from selected ten healthcare facilities of Gujarat, India.

Area	Sites	*P. aeruginosa*	*Pseudomonas* Species	*Acinetobacter* Species	*E. coli*	*Klebsiella* Species	*Staphylococcus* Species
Maternity Ward/ Labor Room/ Delivery Room	Bed	0	4	3	0	1	1
	Mop	7	3	0	0	0	0
	Sink-tap	0	1	1	1	1	2
	Instruments	0	0	0	0	0	0
	Buckets	5	1	0	0	0	0
	NBCU side area	0	0	0	0	0	0
	Table	0	0	0	0	0	1
	Toilet	0	0	0	0	0	0
General Ward	Sink	0	0	0	0	0	0
	Water point	0	0	0	0	0	0
	Staff toilet	0	0	0	0	0	0
	Mop	1	1	0	1	0	0
	Tap	1	0	0	0	0	0

**Table 2 healthcare-10-00648-t002:** Antibiotic resistance pattern among isolated pathogens from different environment surfaces of maternity wards, labor rooms, and general wards of the selected healthcare facilities of Gujarat, India.

Area	Sites	*P. aeruginosa* (%)			*Pseudomonas* Species (%)			*Acinetobacter* Species (%)			*E. coli* (%)			*Klebsiella* Species (%)			*Staphylococcus* Species (%)		
		**R**	**I**	**S**	**R**	**I**	**S**	**R**	**I**	**S**	**R**	**I**	**S**	**R**	**I**	**S**	**R**	**I**	**S**
Maternity Ward/ Labor Room/ Delivery Room	Bed	-	-	-	28.6	57.1	14.3	-	-	100	-	-	-	18.2	4.5	77.3	50	43	6.25
	Mop	50	15.3	34.7	38.1	11.9	50	-	-	-	-	-	-	-	-	-	-	-	-
	Sink-tap	-	-	-	21.4	14.3	64.3	-	-	100	-	-	100	18.2	-	81.8	43.8	3.1	53.1
	Bucket	47.1	20	32.9	28.6	14.3	57.1	-	-	-	-	-	-	-	-	-	-	-	-
	Table	-	-	-	-	-	-	-	-	-	-	-	-	-	-	-	81.3	-	18.6
General Ward	Mop	-	-	100	-	-	100	-	-	-	-	-	100	-	-	-	-	-	-
	Tap	-	-	100	-	-	-	-	-	-	-	-	-	-	-	-	-	-	-

I: Intermediate; R: Resistant; S: Susceptible.

## Data Availability

Data from this study is available at the Indian Institute of Public Health Gandhinagar (IIPHG), Gujarat, India. Researchers who meet the criteria for access to confidential data are encouraged to approach Deepak Saxena, IIPHG, Opp. Air Force Head Quarters, Nr. Lekawada, Gandhinagar 382042, Gujarat, India. Email: ddeepak72@iiphg.org.

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
