# Peer review of "Evidence of Antimicrobial Resistance from Maternity Units and Labor Rooms: A Water, Sanitation, and Hygiene (WASH) Study from Gujarat, India"

_healthcare, 2022, doi:10.3390/healthcare10040648_

Round 1
Reviewer 1 Report
Although the manuscript deals with the important problem of nosocomial infections in maternity wards, it is practically purely descriptive, informative and resembles a report from sanitary-epidemiological stations, which can be easily obtained from the Ministry of Health data. It is advisable to expand the discussion at least to include causal relationships and possible health care guidance from the results. The original literature is quite old. More recent literature is advisable. Row 168 - literature references indicated.
Author Response
Reply to the reviewer’s comment (Reviewer-1)
Journal: Healthcare (ISSN 2227-9032)
Manuscript ID: healthcare-1613998
Type: Article
Title: Evidence of Antimicrobial Resistance from maternity units and labor rooms- A Water, Sanitation and Hygiene (WASH) study from Gujarat, India
Comment: Although the manuscript deals with the important problem of nosocomial infections in maternity wards, it is practically purely descriptive, informative and resembles a report from sanitary-epidemiological stations, which can be easily obtained from the Ministry of Health data. It is advisable to expand the discussion at least to include causal relationships and possible health care guidance from the results. The original literature is quite old. More recent literature is advisable. Row 168 - literature references indicated.
Response: Thanks for the critical comment. We do agree with all of your suggestions and thus we have attempted to revise our manuscript and have added more recent literature stating the importance of the study. In the context of India, where there is no routine data available on the AMR except few selected larger laboratories, this kind of study generates potential evidence on the importance of having microbiological surveillance for understanding the burden of nosocomial infection. This study is also important as such kind of information is not openly available at the ministerial sites and furthermore, such kinds of approaches are not in practice. Thus, we intend to highlight the importance of microbiological surveillance in routine practices. As suggested, we have also discussed the same critically in the discussion chapter of the manuscript.
Changes: Overall manuscript, especially the introduction and discussion chapters are revised. Major changes done in Page 1-2, Line 29-88 & Page: 6-8, Line: 206-333.
Reviewer 2 Report
Overall, this paper presents valuable information about the type of bacteria present in maternity units and the resistance of these bacteria to antibiotics. The introduction provides some good background and context but could be significantly improved with more appropriate use of language. It would also be beneficial to provide some background around hospital acquired infections globally, and locally as well as the consequences for these and how they effect health service delivery (mortality, cost, LOS etc..). Including more references to other related work would be useful.
The methods section is generally well described, some editing of language would improve the readability. Also Lines 73-80 look as though they have been copied and pasted from elsewhere - formatting will need to be corrected. It would also be useful to know why a Roundtable was held to discuss study sites - why did such a rigorous discussion need to happen? Was it based on previous studies?
The results section is clear and aligns well with the aims of the research. Table 1 is easy to read but Table 2 could be improved.
The discussion is fair - it could be improved by more high level discussion, that is what do the results mean in terms of care, service delivery, patient outcomes? There are a lot of gaps. Similarly, the conclusion would benefit from a statement that shows how your findings will benefit the intended participants? What can be done with this information? Locally, globally?
Author Response
Reply to the reviewer’s comment (Reviewer-2)
Journal: Healthcare (ISSN 2227-9032)
Manuscript ID: healthcare-1613998
Type: Article
Title: Evidence of Antimicrobial Resistance from maternity units and labor rooms- A Water, Sanitation and Hygiene (WASH) study from Gujarat, India
Comment-1: Overall, this paper presents valuable information about the type of bacteria present in maternity units and the resistance of these bacteria to antibiotics. The introduction provides some good background and context but could be significantly improved with more appropriate use of language. It would also be beneficial to provide some background around hospital acquired infections globally, and locally as well as the consequences for these and how they effect health service delivery (mortality, cost, LOS etc..). Including more references to other related work would be useful.
Response: Thanks for your valuable comments. As suggested, we have now added more recent literature to the introduction chapter to build the context and state the importance of the study. Changes: Page 1-2, Line 29-88
Comment-2: The methods section is generally well described, some editing of language would improve the readability. Also Lines 73-80 look as though they have been copied and pasted from elsewhere - formatting will need to be corrected. It would also be useful to know why a Roundtable was held to discuss study sites - why did such a rigorous discussion need to happen? Was it based on previous studies?
Response: Thanks for pointing out this method. Yes, this was done prior to the data collection primarily for the two reasons- There were no structured guidelines which are the potential areas from where the swab samples to be collected and as this is a research study keeping the financial constraint, researchers wanted to have a consensus on number of swab samples and sites that are more relevant to the local context. To build the consensus as per the finding of the previous research which are more valid to the local context.
Changes: Page 2-3 Line 90-146
Comment-3: The results section is clear and aligns well with the aims of the research. Table 1 is easy to read but Table 2 could be improved.
Response: Thanks for the positive comment. We have included the foot notes to the table to and attempted to describe the same in the result section.
Changes: Page 3-5, Line 128-204
Comment-4: The discussion is fair - it could be improved by more high level discussion, that is what do the results mean in terms of care, service delivery, patient outcomes? There are a lot of gaps. Similarly, the conclusion would benefit from a statement that shows how your findings will benefit the intended participants? What can be done with this information? Locally, globally?
Response: Thank you for the suggestion. Now the discussion section is reconstructed and revised as per the suggestion.
Changes: Page: 6-8, Line: 206-333
Round 2
Reviewer 2 Report
Thank you to the authors for addressing the previous feedback. The manuscript has significantly improved. The background and context is well described and the use of supporting literature is very good. The discussion in particular has improved and is more inclusive of higher level thinking and conclusions that support your aims. Well done.